# Leukocyte Trafficking via Lymphatic Vessels in Atherosclerosis

**DOI:** 10.3390/cells10061344

**Published:** 2021-05-29

**Authors:** Kim Pin Yeo, Hwee Ying Lim, Veronique Angeli

**Affiliations:** Immunology Translational Research Programme, Department of Microbiology and Immunology, Yong Loo Lin School of Medicine and Immunology Programme, Life Sciences Institute, National University of Singapore, Singapore 119077, Singapore; kimpinyeo@yahoo.com (K.P.Y.); miclhy@nus.edu.sg (H.Y.L.)

**Keywords:** lymphatic vessel, atherosclerosis, adventitia, lymphangiogenesis, immune cells, cholesterol, inflammation

## Abstract

In recent years, lymphatic vessels have received increasing attention and our understanding of their development and functional roles in health and diseases has greatly improved. It has become clear that lymphatic vessels are critically involved in acute and chronic inflammation and its resolution by supporting the transport of immune cells, fluid, and macromolecules. As we will discuss in this review, the involvement of lymphatic vessels has been uncovered in atherosclerosis, a chronic inflammatory disease of medium- and large-sized arteries causing deadly cardiovascular complications worldwide. The progression of atherosclerosis is associated with morphological and functional alterations in lymphatic vessels draining the diseased artery. These defects in the lymphatic vasculature impact the inflammatory response in atherosclerosis by affecting immune cell trafficking, lymphoid neogenesis, and clearance of macromolecules in the arterial wall. Based on these new findings, we propose that targeting lymphatic function could be considered in conjunction with existing drugs as a treatment option for atherosclerosis.

## 1. Introduction

The lymphatic system is part of the human circulatory system and the immune system. It is composed of an extensive branched network of lymphatic vessels that are connected to lymph nodes (LNs). Unlike the closed blood circulatory system, lymphatic vasculature functions unidirectionally. Extravasated fluid called lymph, macromolecules, and leukocytes enter the blind-ended initial lymphatic vessels (also called lymphatic capillaries) and are transported towards the pre-collecting vessels that converge to the larger collecting vessels, pass through chains of LNs, before returning to the blood circulation through the thoracic duct [1,2]. Lymphatic vessels are present in all vascularized tissues including skin and most internal organs except bone marrow [1]. Hence, each tissue and organ is typically connected to the draining LNs.

Initial lymphatic vessels, the absorptive part of the lymphatic vessels, are lined by a single layer of oak leaf-shaped lymphatic endothelial cells (LECs) which are not covered by pericytes or smooth muscle cells (SMCs) [3]. Initial lymphatics have discontinuous button-like junctions and are anchored to the extracellular matrix by elastic fibers or anchoring filaments which prevent the vessels from collapsing during high interstitial pressure [4]. These structural features render the initial lymphatics highly permeable, which enables the optimal entry of large macromolecules, pathogens, and immune cells from the interstitial tissues [1]. In contrast, collecting lymphatic vessels exhibit continuous zipper-like cell–cell junctions, valves, and are ensheathed with a basement membrane and SMCs [4,5]. The intrinsic contractility of SMCs and the extrinsic contraction of surrounding skeletal muscles and arterial pulsations are necessary for lymph propulsion through the vessel while the valves prevent lymph backflow [6,7,8].

Lymphatic vessels have been shown to play essential roles in immune surveillance by, for example, supporting antigen and immune cell transport, dietary fat absorption in the gastrointestinal tract, and maintenance of tissue fluid balance [1,9,10,11]. Until recently, the major symptomatic pathological condition associated with defects in lymphatic functions was lymphedema. However, lymphatic vessels received increasing attention in the last years and novel, active, and functional roles of the lymphatic vasculature in health and diseases have been discovered. As such, the list of human diseases or disorders associated with defects in lymphatic functions has grown larger and includes Crohn’s disease, neurological disorders, eye diseases such as glaucoma, inflammation, and cardiovascular diseases [12].

In this review, we focus on the functional roles of lymphatic vessels draining medium- and large-sized arteries in atherosclerosis, a chronic inflammatory disease of arteries. We first provide a brief overview of atherosclerosis pathogenesis and then discuss the normal distribution of lymphatic vessels in arteries and their morphological and functional alterations in response to atherosclerosis. Finally, we review recent studies revealing the important contribution of lymphatic vasculature in atherosclerosis progression through the control of immune cell trafficking and lipids.

## 2. Atherosclerosis: A Chronic Inflammatory Disease of the Arterial Wall

### 2.1. Intimal Atherosclerotic Plaque Development and Progression

Atherosclerosis is a chronic inflammatory disease of medium-sized to large arteries, and the associated cardiovascular complications including myocardial infarction and stroke represent leading causes of death worldwide. The arterial wall consists of three concentric tissue layers namely: the tunica intima (inner layer) lined by a monolayer of endothelial cells, the tunica media composed of SMCs, elastin, and collagen, and the adventitia, the outer layer formed of fibroblasts, collagen, nerve endings, immune cells, and blood and lymphatic vessels (Figure 1). Atherosclerosis develops gradually and silently over decades, continuously progressing from early lesions characterized by macrophages loaded with cholesterol ester to advanced lesions with more complex cellular composition, lipid pools, formation of fibrous cap, and necrotic core [13].

According to the traditional inside-out theory of atherosclerotic lesion development, the disease is initiated from the luminal side of the artery and involves several orchestrated mechanisms triggered by endothelial activation and subendothelial retention of plasma apolipoprotein in the intima, including monocyte adhesion on the luminal surface, endothelial dysfunction, leukocyte accumulation in the subendothelial space, and subsequent inflammatory responses [14,15]. The extracellular modifications of lipoproteins such as oxidation induce the local innate and adaptive immune responses in the intima through the activation of inflammatory cells, which in turn respond by producing pro-inflammatory cytokines and chemokines [16]. The latter promote further recruitment of various circulating leukocytes and local activation of endothelial cells. The inflammatory cells that infiltrate the developing atherosclerotic lesions include monocytes and monocyte-derived macrophages, dendritic cells, various subsets of T and B lymphocytes, mast cells, and, in the more advanced, often thrombotic lesions, neutrophils [17,18,19,20,21,22,23,24,25]. The intimal monocyte-derived cells also contribute to the modification of the apolipoprotein B-containing lipoproteins through the secretion of lipoprotein-modifying enzymes or agents [26,27,28]. Uptake of modified lipoproteins or of cholesterol crystals by macrophages, dendritic cells, and vascular SMC leads to the accumulation of cytoplasmic cholesteryl ester droplets leading to the formation of “foamy” cells, a histological hallmark of atherogenesis. Since excess of un-esterified or free cholesterol is toxic to cells, foam cell formation involving esterification and storage of cholesterol as cholesteryl ester droplets can be considered as a beneficial process in the early atherosclerotic lesions. During disease progression, continuous recruitment of monocytes, deposition of cholesterol crystals, migration of medial SMCs to the intima, and undesirable immunity against cholesterol-associated apolipoproteins trigger and sustain chronic inflammation. In advanced lesions, foams cells die and release their contents into the extracellular microenvironment and contribute further to inflammation, formation of lipid-rich necrotic core, and plaque instability. Unstable plaques are also characterized by the thinning of fibrous cap made of collagen, elastin, and proteoglycan, which in turn may promote its fracture and allow blood coagulation components to interact with plaque contents such as tissue factor, triggering thrombosis and subsequent clinical complications associated with atherosclerosis.

Taken together, infiltration and accumulation of plasma lipoproteins and pro-inflammatory leukocytes in the intima represent the driving force of atherosclerotic lesion initiation, progression, and clinical complications. It is thus not surprising that targeting lipoprotein and inflammatory pathways has been the main approaches in managing atherosclerosis. Statins, 3-Hydroxy-methylglutaryl coenzyme A (HMG Co-A) reductase inhibitors, have been the standard treatment for atherosclerosis-associated diseases [29]. More recently, a novel medicine to reduce LDL-cholesterol has been approved by Food and Drug Administration targeting and inactivating the proprotein convertase subtilsin-kexin type 9 (PCSK9). PCSK9 is a hepatic protease that attaches and internalizes low density lipoprotein receptors into lysosomes, hence promoting their destruction. By preventing LDL receptor destruction using monoclonal antibodies against PCSK9, LDL cholesterol levels can be lowered 50–60% above that achieved by statin therapy alone [30,31]. In contrast to the lack of success with previous anti-inflammatory drugs [32,33], a monoclonal antibody against the pro-inflammatory interleukin 1 beta studied in the Canakinumab Anti-Inflammatory Thrombosis Outcomes Study (CANTOS) trial has been shown to reduce cardiovascular events in patients with recent acute coronary syndrome and persistent levels of C-Reactive Protein [34]. However, this was associated with a significant increased rate of fatal infection [34]. Nevertheless, this landmark study provides the first evidence that targeting inflammation is also a promising approach in decreasing atherosclerosis-related risks.

### 2.2. Adventitial Inflammatory and Immune Response in Atherosclerosis

Although atherosclerotic lesions predominantly localize in the intima, atherosclerosis affects the structure and the functions of all three layers of the arterial wall. Notably, emerging evidence recognizes the adventitia as an active participant in the development and progression of atherosclerotic lesions which have led to a new paradigm of an outside-in theory, in which vascular inflammation is initiated in the adventitia and progressed inwards toward the intima [35,36,37]. A normally functioning adventitia is crucial for the homeostasis of the entire arterial wall, since the removal of adventitia in animal models [38] induces intimal hyperplasia, involving both SMCs and macrophages [39], that regresses with adventitia regrowth. Adventitia is a highly responsive tissue that can regenerate quickly in response to experimental stripping [38]. Increasing evidence reveal that several events taking place in the adventitia have consequences on biological processes in the intima and media. For example, infiltration of immune cells has been demonstrated to correlate with the development of atherosclerotic lesions in humans and mice [40,41].

It is increasingly recognized that intimal inflammation is paralleled over time by a distinct inflammatory reaction to adjacent adventitia (Figure 1). Various immune cells reside in normal, noninflamed artery adventitia, including macrophages, dendritic cells, T cells, and, to a lesser extent, B cells [42]. Although monocytic cell infiltration has been traditionally considered as an event mainly occurring in the intima during the early atheromatous process, inflammatory cells can also infiltrate in the adventitia adjacent to atherosclerotic plaques. Adventitial mononuclear cell infiltration associated with human atheromatous plaques was reported in coronary arteries by Gerlis in 1956 [43] and by Schwartz and Mitchell in 1962 [44]. A similar adventitial response was further documented in the human aorta by Parums and Ramshaw [45]. In the atherosclerotic mouse model lacking apolipoprotein E (*Apoe^−/−^*), the number of macrophages, T cells, and microvessels was significantly higher in the adventitia of the abdominal aorta and these numbers positively correlated with plaque formation and collagen deposition [37]. A recent study in *Apoe^−/−^* mice employing single cell RNA sequencing to further characterize the cellular landscape of the adventitia at a single-cell resolution confirmed that this tissue harbors inflammatory cells and uncovered subpopulations of lymphocytes, macrophages, and several rare populations such as lymphatic endothelial cells and neuronal cells [46]. Interestingly, adventitial non-clustered T cells predominate in young *Apoe^−/−^* mice, whereas clusters containing T and B cells as well as lymphoid-like structures emerge in non-resolving inflammation in aortic adventitia of older mice, preferentially in the abdominal aorta [37]. T cells also reside in the adventitia of normal healthy aortas of C57BL/6 mice [47] and *Apoe^−/−^* mice [37,47].

The progressive organization of cellular infiltrates within non-lymphoid organs into functional ectopic germinal centers or also known as tertiary lymphoid organs (TLOs) is a process called lymphoid neogenesis [48]. Lymphoid neogenesis has been observed in chronic infectious diseases, autoimmune diseases, and chronic rejection of solid organ transplants [49,50]. Adventitial TLOs are well-organized structures with areas of T cells, B cells, antigen-presenting cells, follicular dendritic cells, lymphatic vessels, and high endothelial venules [51,52,53] (Figure 2). Early atherosclerosis is accompanied with significant T-cell infiltrates in the intima whereas T-cell numbers in plaques markedly reduces over time and markedly increases in the adventitia during aging in mice [37]. Unlike T cells, B cells are absent in the normal aorta and in atherosclerotic lesions, but form clusters during intermediate stages of adventitial TLOs neogenesis and ectopic germinal centers containing follicular dendritic cells with proliferating B-cell centroblasts in more advanced stages of adventitial TLOs [37]. Human adventitial TLOs have not yet been well characterized, although adventitial aggregates containing T cells, B cells, and follicular dendritic cells were reported in patients with atherosclerotic abdominal aneurysms [41]. Adventitial leukocyte infiltrates are associated with clinically significant stages of atherosclerosis, and are more abundant in ruptured compared with non-ruptured plaques in patients with abdominal aorta atherosclerosis [54]. Altogether, these findings provide evidence that adaptive T and B cell responses take place within the diseased arterial adventitia. Although the specific antigens inducing these responses remain to be identified, stress proteins (heat shock proteins), modified lipoproteins, and other surface antigens have been implicated [53,55].

While adventitial TLOs are associated with atherosclerotic lesions in the intima, suggesting a crosstalk between the intimal lesions and adventitial TLOs via the media, the formation of advanced plaques is not sufficient to promote the formation of adventitial TLOs in *Apoe^−/−^* mice. Indeed, TLOs are rarely observed in the adventitia of the aortic arch and its branches, although atherosclerosis is the most advanced. Although locally released soluble mediators such as lymphoid chemokines and lymphotoxin have been involved in the formation of adventitial TLOs in atherosclerotic mice [56], the triggers for the cascade of events leading to lymphoid neogenesis in adventitia of atherosclerotic arteries remain to be identified [57,58].

## 3. Adventitial Blood and Lymphatic Vasculature

### 3.1. Adventitial Neovascularization of Vasa Vasorum in Atherosclerosis

Adventitial small blood vessels also named vasa vasorum are small arteries reaching the vascular wall either from the abluminal surface (vasa vasorum externa) or from the luminal surfaces (vasa vasorum interna) and branching out to the outer media [59]. Venous vasa vasorum drain a network of capillaries/venules lining the outer media of veins near the arteries. The major function of vasa vasorum is to transport nutrients and oxygen to arterial and venous walls and to clear “waste” products released from cells in the wall or brought in by diffusion through the arterial or venous endothelium. A study in which intercostal ligations of arteries that interrupt blood flow through the vasa vasorum, induced localized necrosis and cell loss in the intima and medica layers, illustrating the role of vasa vasorum in the nourishment of the aortic wall [60]. Moreover, the vasa vasorum have received considerable interest for more than a century because of their possible implication in atherogenesis [38,61,62,63,64] and in response to risk factors for atherosclerosis, such as hypercholesterolemia [65,66] and hypertension [67,68,69]. Indeed, experimental manipulation of the vasa vasorum can lead to atherosclerotic changes of the intima [38,61,64,70]. Numerous autopsy-based studies, together with experimental studies in animal model of atherosclerosis such as *Apoe^−/−^* mice, revealed that atherosclerotic lesion formation is associated with neovascularization of the vasa vasorum [71,72] and that this process occurs in the neointima, increases with, and determines plaque severity [73]. Notably, the histological analysis of coronary arteries by Kumamoto et al. revealed the appearance of new blood capillaries in the atherosclerotic plaque that arose mainly from the adventitia and rarely from the lumen [73]. In humans, neovascularization emerges largely in the shoulder of the complicated plaque, at the interface between the core, the cap, and the media.

The study by Moreno et al. in the human aorta showed that neovascularization is augmented in human atherosclerotic plaques with inflammation, intraplaque hemorrhage, thin fibrous cap, and in ruptured plaques [74]. This finding was further supported by the more recent study of Langheinrich et al. showing the association among different advanced atherosclerotic lesions, adventitial vasa vasorum neovascularization, and adventitial inflammation [72]. Moreover, inhibition of angiogenesis in *Apoe^−/−^* mice not only reduced atherosclerotic lesion neovascularization, but also lesion growth [75]. Infiltration of lipids into the intima or the media together with oxidative stress may generate angiogenic stimulus promoting the proliferation of vasa vasorum. Notably, the inhibition of angiogenesis leads to decreased macrophages in the plaque and in the adventitia, especially around the vasa vasorum [76]. These data strongly suggest that the neovascularization of the vasa vasorum may serve as a channel for entry of macrophages and inflammatory mediators that may potentially support the progression of atherosclerosis [76]. Since pathological neovascularization is often accompanied by increased endothelial permeability [77], pro-atherogenic leukocytes and soluble pro-atherogenic agents may enter the arterial wall more easily through ruptures and/or leaky vasa vasorum, further promoting the progression of atherosclerosis [78].

In conclusion, during the development of atherosclerosis, vasa vasorum from the tunica adventitia may penetrate the thickened media to provide nutrition and oxygenation. Moreover, neovascularization of vasa vasorum together with the endothelial dysfunction of these nutritive vessels may significantly contribute to atherosclerosis by supporting the infiltration of inflammatory cells in the intima and the adventitia.

### 3.2. Adventitial Lymphatic Vessels

#### 3.2.1. Lymphatic Vessel Distribution in Normal Artery

The existence of lymphatics in the arterial wall was studied by G. Hoggan and colleagues more than one hundred years ago when specific lymphatic markers were not available [79]. When Hoggan et al. examined lymphatic vessels of the larger blood vessels of goat and horse with light microscopy, they did not find lymphatics in the intima and media, but a plexus of lymphatic vessels was observed lying in the adventitia of larger arteries. They also observed that the network of lymphatic vessels joined at one point and passed through the adventitia to join larger lymphatic trunks which led the lymph away [79]. Subsequently, in the mid 20th century, R.A Johnson emphasized the similar anatomical observation of adventitial lymphatic vessels and these lymphatics did not penetrate the media from the adventitia in the pig, dog, and human [80]. Lymphatic vessels in the carotid and thoracic aorta were studied in rabbits and guinea pigs using transmission electron and light microscopy in the early 1990s [81]. Lymphatic vessels consistently occupy the adventitial and loose connective tissue of the peri-adventitial area and appear to be larger than blood capillaries, interconnected and wrapped around larger-caliber arteries. Anatomical relationship of lymphatics to the aortic wall in small animal such as the Wistar rat was then reported [82]. Using Lymphatic Vessel Endothelial Receptor 1 (LYVE-1) and podoplanin antibody, Drozdz et al. further substantiated the presence of adventitial lymphatic vessels in several parts of aorta and arteries in human, including internal carotid arteries, abdominal aortas, and iliac arteries [83]. In contrast to these studies, Eliska et al. and Nakano et al. did not detect lymphatic vessels in normal aorta [84,85].

As most mechanistic insights depend on in vivo studies using smaller mouse model, it is critical to know the exact topography of aortic lymphatic vessels in a mouse model. Using histological and imaging analysis, characterization and topographic distribution of the lymphatic vessel network in the normal mouse aorta were recently illustrated [86]. Blind-ended lymphatic vessels expressing LYVE-1, vascular endothelial growth receptor–3 (VEGFR-3), podoplanin, and prox-1 were found residing in the collagen I-rich aortic adventitia and peri-adventitial tissues of the aortic sinus (Figure 2), thoracic aorta, and abdominal aorta, while aortic arch is destitute of lymphatic vessels. Lymphatic vessels ran alongside the thoracic and abdominal aorta, having one plexus anatomically closer to the vein while another weaving in and out the adjacent periadventitial adipose tissues and encircling the arterial branches [86].

Overall, an irregular plexus of lymphatic vessels was consistently found in the adventitia and peri-adventitial area, but not in the arterial intima and media, showing a conserved presence and anatomical location in both small and large mammals including humans.

#### 3.2.2. Morphological and Density Alterations in Lymphatic Vessels during Atherosclerosis

Lymphatic vessels are highly dynamic structures that intimately interact with their surrounding tissue microenvironment. Recent research advances in lymphatic biology have revealed that various internal and external stimuli can affect and modulate the growth, structure, and function of lymphatic vessels. Lymphangiogenesis and enlarged lymphatic vessels are associated with chronic inflammation, such as psoriasis [87] or chronic airway inflammation [88]. As early as 1969, a plexus of lymphatic network was reported to be present in adventitia but not media of human aorta containing atheroma by injection of colored dye [80]. Using lymphatic-specific markers, Drozdz et al. subsequently confirmed the early observation and showed a correlation between intimal thickness and adventitial lymphatic density in human internal carotid artery, abdominal aorta, and iliac arteries [83]. The author further speculated that inflammatory environment contributed to lymphangiogenesis. The presence of adventitial lymphatic vessels was not only observed in the human coronary artery, but also detected within the atherosclerotic lesions [89]. Contrary to these findings, lymphangiogenesis is rarely observed in human coronary arteries and carotid endarterectomy specimens despite a high endogenous expression of vascular endothelial growth factor-C (VEGF-C) in atherosclerotic plaque [85,90].

Similar expansion of adventitial lymphatic network was also observed in hypercholesterolemic *Apoe^−/−^* mice [91,92]. The temporal and spatial assessment of lymphangiogenesis was carefully studied at various regions of the aortae in *Apoe^−/−^* and *Ldlr^−^^/−^* mice [86]. Adventitial lymphangiogenesis develops at later stages of plaque development and dilated lymphatic vessels were more numerous in advanced plaques. Reducing hypercholesterolemia by ezetimibe decreases the expansion of lymphatic vessels and improves their morphological changes in *Apoe^−/−^* mice [86]. Furthermore, lymphangiogenesis is enhanced through treatment with VEGFC-156S (VEGF-C protein with a substitution of Cys^156^ by a serine residue), but reduced through VEGFR-3 inhibition, suggesting that hypercholesterolemia may trigger lymphangiogenesis in *Apoe^−/−^* mice through a mechanism dependent, in part, on VEGF-C [86]. However, manipulating VEGFR-3/VEGF-C signaling did not affect atherosclerotic plaque size and plasma cholesterol level. Oxidized LDL may also alter the growth of adventitial lymphatic vessels during atherosclerosis by acting on lymphatic endothelial cells through a CD36-dependent mechanism [93].

It is worth mentioning that lymphangiogenesis and angiogenesis co-exist during atherosclerosis progression and regression. Both vasculature networks are intimately intertwined, even though there is no connection between the two systems. The kinetic of angiogenesis also differs from lymphangiogenesis, with angiogenesis preceding and surpassing lymphangiogenesis during disease progression, which results in the imbalance of the ratio of blood to lymphatic vessel numbers [86].

Furthermore, a hypertrophy in aorta draining iliac LNs was constantly observed in hypercholesterolemic mice [86]. Changes in the aorta draining LNs may result from increased fluid burden, expansion of lymphatic sinuses and fibroblastic reticular cell network, proliferation of lymphocytes, and failure in lymphocytes egress [94]. Although lymphangiogenesis was not noticeable in the skin of *Apoe^−/−^* mice, it was apparent in the hypertrophic skin draining LNs [94]. Microscopic examination of skin draining LNs revealed that the lymphatic networks were notably dilated with open lumens as compared to those of the wild-type LN in a more partially collapsed state. Given the pronounced increase and changes in the adventitial lymphatic network, it is not surprising that similar LN lymphatic remodeling also occurs in the iliac LNs.

## 4. Functional Roles of Adventitial Lymphatic Vessels in Atherosclerosis

After reviewing the alterations of the lymphatic vasculature observed during atherosclerosis (Figure 1), we will discuss below the possible implications of lymphatic vessels and their functional defects in the inflammatory response associated with atherosclerosis.

An artery is formed by three layers, namely: the tunica intima (inner layer) lined by a monolayer of endothelial cells, the tunica media composed of SMCs, elastin and collagen, and the adventitia, the outer layer formed of fibroblast, collagen, nerve endings, immune cells, and blood and lymphatic vessels. Periadventitial adipose tissue is also distributed around arteries. Atherosclerosis is characterized by the accumulation of plaques comprising cholesterol, inflammatory soluble factors, immune cells in the intima, and inflammation in adventitia that can be associated with the formation of adventitia tertiary lymphoid organ (ATLOs) when the disease is more advanced. Similar to lymphatic vessels in other tissues, initial lymphatic vessels in the adventitia converge into larger collecting vessels exhibiting SMC and valves which transport lymph into draining lymph nodes. Studies have shown that atherosclerosis is associated with angiogenesis and lymphangiogenesis. Notably, in the abdominal aorta, initial lymphatic vessels appeared more dilated and lymphatic drainage of macromolecules from the adventitia into the draining lymph node is severely compromised during atherosclerosis. Alterations in collecting vessels such as poor SMC coverage, abnormal valve, and accumulation of extracellular matrix, such as collagen, may account for the impaired lymphatic drainage. These functional defects in adventitial lymphatic vessels may participate in the retention of immune cells, inflammatory cytokines, and lipids in the arterial wall, which in turn favor the progression of atherosclerotic plaque, inflammation, and ATLOs. Restoring lymphatic drainage in the arterial wall may thus be a promising strategy to treat atherosclerosis.

### 4.1. Emigration of Immune Cells to Lymph Nodes

Lymphatic vessels, together with their interconnected LNs, are essential for adaptive immune responses and the clearance of inflammatory cytokines, antigens, and cells that may cause harm if not removed from tissues. Through integration with afferent lymphatic vessels, LNs receive antigens, antigen presenting cells such as dendritic cells, and macrophages and lymphocytes for the induction of immune responses [95]. Whereas the LN subcapsular sinus supports the entry of cells and molecules into the parenchyma of the LN [96], the medullary sinuses localized in the hilum of the LN mediate their exit from the LN. Lymphangiogenesis occurring in the inflamed tissue supports the transport of lymph containing activated dendritic cells, inflammatory cytokines, or antigens into draining LNs to prime an immunological response. Inflammation also stimulates the expression of the chemokine CCL21 [97] and integrins such as intercellular adhesion molecule-1 (ICAM-1), vascular adhesion molecule-1 (VCAM-1), and E-selectin [98] on the lymphatic endothelium. CCR7–CCL21 signaling facilitates the entry of DCs [97,98,99] and T lymphocytes [100,101] into afferent lymphatic vessels during inflammation. ICAM-1 and VCAM-1 also supports the trafficking of DCs into afferent inflamed lymphatic vessels [97]. Since the hallmark of atherosclerosis is the accumulation of immune cells including monocyte, dendritic cell, macrophage, and T lymphocyte in the aortic wall, lymphatic vessels might have a beneficial role in promoting the migratory egress of these immune cells to the LNs to bring about resolution of the plaque and adventitial inflammatory response.

#### 4.1.1. Emigration of Monocyte, Macrophage, and Dendritic Cell

Based on the recent literature, several processes have been proposed to account for the accumulation of monocyte-derived cells in atherosclerotic plaque, namely: increased monocyte recruitment into the plaques, alterations in proliferation and/or survival of monocyte-derived cells in plaques, and inability of macrophage and dendritic cells to emigrate from plaques to LNs via adventitial lymphatic vessels [102,103]. Macrophages are normally removed from sites of resolving inflammation by migration through lymphatics to LNs [104]. In a surgical model of plaque regression which involves surgical transfer of a plaque-bearing aortic segment from a mouse with atherosclerosis to another wild-type recipient mouse with low levels of circulating cholesterol [105], rapid removal of plaque foam cells correlates with the onset of their emigration to draining LNs [106]. Moreover, this migration seems to depend upon CCR7 [107]. In contrast, if the recipient mouse is hypercholesterolemic, then the plaques in the newly transferred portion of the aorta progress and continue to accumulate foam cells which rarely emigrate to LNs [106]. This study suggests that emigration of monocyte-derived cells from the plaques via lymphatics is a feature of atherosclerotic plaque regression, but not progression. However, this conclusion may depend on the model of plaque regression used in the study. Indeed, in a model of plaque regression which does not involve surgery but the treatment of *Apoe^−/−^* mice with a viral encoding apoE which lowers plasma cholesterol and raises high-density lipoprotein, the reduction in plaque macrophage content did not result from migratory egress and did not require CCR7 [108]. Although several explanations may account for the differences observed between these two regression models, one possibility is that the surgery may have affected the barrier function to leukocyte trafficking of the elasticized medial SMC layers [109]. Indeed, leukocytes are rarely found within the medial layer in *Apoe^−/−^* mice, unlike in human arteries burdened with atherosclerotic plaques where phagocytes can be observed in the media at a greater percentage than in normal arteries [110]. Notably, increased cell infiltration in the media was noticed together with an exacerbated adventitial inflammation in the surgical model [106]. Although this latest study in non-surgical regression model suggests that emigration of monocyte-derived cells via lymphatics may not be the major mechanism for macrophage removal from atherosclerotic plaque [108], promoting migratory egress of immune cells through lymphatic vessels remains a promising additive strategy to prevent atherosclerosis development or progression. This is further supported by our recent findings. We demonstrated that ligation of lymphatic vessels draining the abdominal aorta of *Apoe^−/−^* mice with negligible atherosclerotic plaques promotes the intimal accumulation of macrophages, collagen, and SMCs as well as macrophage and dendritic cell infiltration in adventitia layer [86]. Moreover, although lymphatic ligation did not promote atherosclerotic plaque formation in normocholesterolemic wild-type mice, it was sufficient to induce adventitial accumulation of macrophage and dendritic cells [86]. Altogether, these data indicate that lymph stasis in the aortic wall induced by lymphatic ligation can promote the development of atherosclerotic plaque. Since we showed that aortic lymphatic drainage in *Apoe^−/−^* is significantly compromised compared to wild-type mice [86], this suggests that defects in aortic lymphatic drainage contributes to atherosclerosis by preventing the emigration of monocyte-derived cells, at least from the adventitia (Figure 1).

#### 4.1.2. Emigration of T Lymphocyte

In addition to the control of dendritic cell and macrophage trafficking, lymphatic vessels are also critical for T cell migration from tissue into LNs and T cell egress from LNs into efferent collecting lymphatic vessels. Among the chemokines expressed by lymphatic endothelium that attract T cells, CCL21 is essential for the migration of naïve, memory, and T regulatory T cells to LNs [111]. On the other hand, T cell egress is controlled by sphingosine-1-phosphate (S1P)/S1P1 signal axis [112]. Specifically, low levels of S1P in LN parenchyma guide T cells expressing decreased CCR7-retential signal into medullary and cortical sinuses where S1P levels are high, thus facilitating T cell egress [113]. In the absence of lymphatic endothelial cell–expressed sphingosine kinase 1 and sphingosine kinase 2, sphingosine-1-phosphate is not detectable in LNs, and T cells thus fail to egress [114]. As mentioned above, T lymphocytes also accumulate in intimal atherosclerotic plaques and in adventitia layer. Although the accumulation of T cells in adventitia during atherosclerosis may be driven by adventitial angiogenesis, there are now evidence that lymphangiogenesis and lymphatic drainage may also control this process. To study the involvement of adventitial lymphatic vessels in T cell accumulation during atherosclerosis, Rademakers et al. placed on the carotid of *Apoe^−/−^* mice a semi-constrictive collar to induce atherosclerosis after LN and lymphatic dissection [92]. Interestingly, they demonstrated that blocking lymphatic drainage aggravates atherosclerotic plaque formation and increases intimal and adventitial T cell content. Similar results were observed when adventitial lymphangiogenesis was inhibited by either blocking VEGF-C/VEGFR-3 signaling or silencing locally CXCL12, a novel lymphangiogenic factor [92]. Consistent with this study, we also found an accumulation of T cells in adventitia of abdominal aorta in our model of lymphatic ligation in *Apoe^−/−^* mice [86]. Moreover, the observation that the accumulation of T cells in the adventitia is associated with sluggish aortic lymphatic drainage in *Apoe^−/−^* mice further support the functional role of lymphatic drainage in limiting aortic accumulation of T lymphocytes in adventitia and in turn the inflammatory response in atherosclerosis. Efferent lymphatic emigration of T lymphocytes from LNs may also be compromised during atherosclerosis. Indeed, we showed that T cell egress from skin draining LNs in hypercholesterolemic *Apoe^−/−^* mice is severely impaired, which contributes to LN hypertrophy [94]. The impairment of T cell egress results from the aberrant expansion of cortical and medullary sinuses and imbalance between egress (S1P)/retention (CCL21) signals in the hypertrophic LN. Notably, LN hypertrophy is a common feature of autoimmune and chronic inflammatory diseases. Failure of lymphocytes, especially effector cells, to emigrate from the inflamed LNs in a timely manner may compromise their function and thus immunosurveillance. This might apply to atherosclerosis since aortic LNs also become hypertrophic in *Apoe^−/−^* mice [86], and these mice exhibit impaired priming when immunologically challenged [115] and increased susceptibility to infection [116,117,118,119].

### 4.2. Lymphoid Neogenesis

In an excellent review on the mechanisms triggering lymphoid neogenesis during chronic inflammation, Thaunat O et al. proposed, based on their work in the field of chronic rejection of solid organ transplants, that lymphoid neogenesis is a default response of an immune system resulting from both a chronic local antigenic stimulation and a defect in the lymphatic drainage of the tissue harboring the targeted antigen [120]. According to our current knowledge on the distribution and functionality of adventitial lymphatic vessels, it is likely that this concept may also apply to lymphoid neogenesis taking place in the adventitia adjacent to atherosclerotic plaques. Intriguingly, as discussed above, TLOs are rarely observed in the adventitia of the aortic arch and its branches in *Apoe^−/−^* mice although atherosclerosis is the most advanced [37]. Notably, when mapping lymphatic vessels draining mouse aorta, we consistently noticed that lymphatic vessels are absent or very scarce in the adventitia of the aortic arch compared to abdominal aorta [86]. Moreover, in contrast to vasa vasorum neovascularization, lymphangiogenesis is rare in atherosclerotic plaque. In the abdominal aorta, TLOs and lymphatic vessels coexist in the adventitia during atherosclerosis. However, we demonstrated that despite the apparent lymphangiogenesis, lymphatic drainage at this region of the aorta is significantly reduced in *Apoe^−/−^* mice and promotes atherosclerotic plaque development and inflammation in the adventitia [86] (Figure 1). It is therefore tempting to speculate that insufficient lymphatic drainage of the plaque because of either absence of lymphatic vessels or compromised draining function, contributes to the retention of soluble mediators and immune effectors in the adventitia, thus creating the optimal conditions for TLOs to form (Figure 2).

### 4.3. Drainage of Macromolecules

Given the primary function of lymphatic vessels in draining fluid, protein, and lipid from peripheral tissues back to the blood circulation [12], clearance of intimal lipoproteins and inflammatory cytokines via adventitial lymphatic vessels could be an athero-protective pathway, limiting the accumulation of cholesterol and inflammatory soluble factors in the arterial wall and thus plaque development and inflammation. Recent findings in mouse models have demonstrated that lymphatic vessels are the major route for transport of high-density lipoprotein particles in cholesterol from tissues to bloodstream, an athero-protective process known as reverse cholesterol transport [91,121,122]. In these models, lymphatic insufficiency inhibits by 80% reverse cholesterol transport [91] whereas VEGF-C treatment which promotes lymphangiogenesis or improves lymphatic drainage decreases cholesterol accumulation in tissue and enhances reverse cholesterol transport [121,123]. Therefore, this work supports the essential role of lymphatic vessels in limiting cholesterol accumulation in atherosclerosis (Figure 2). Furthermore, lymphatic vessels may also play an important role in the resolution of inflammation in atherosclerosis by supporting the clearance of inflammatory factors and cells. Indeed, induction of lymphangiogenesis with enhanced fluid drainage has been shown to reduce the severity of inflammation in several diseases including psoriasis, chronic airway inflammation, arthritis, and inflammatory bowel disease [124]. However, the lymphangiogenesis observed in adventitia of atherosclerotic aorta does not translate into increased but decreased lymphatic drainage. Thus, the retention of cytokines, antigens, and cholesterol in the arterial wall could be a consequence of this poor adventitial lymphatic clearance ability which in turn would participate in plaque and adventitia inflammation.

## 5. Is lymphatic Drainage Defective in Atherosclerosis?

Most clinical and pre-clinical studies demonstrate that an increased lymphatic network is necessary for the resolution of inflammation. The fundamental roles that lymphatics plays in fluid clearance from the periphery make lymphangiogenesis seemingly obvious as a mechanism to alleviate locally inflamed environments by increasing lymph efflux. The lymph contains all transportable tissue macromolecules including cytokines, tissue fragments, hormones, and antigens and is carried along larger collecting lymphatics to the draining LNs. Obstruction of the lymphatics along this path exacerbates unfavorable accumulation of macromolecules, which subsequently leads to architectural tissue changes, oedema, and infiltration of immune cells in the peripheral tissues [125,126]. It is, however, unclear if the expansion of the lymphatic vasculature during inflammation correlates with its function: is it a mere response to inflammation, a component of pathology, or an attempt to resolve inflammation?

The link between adventitial lymphangiogenesis and atherosclerosis has been postulated for more than a few decades ago, but only a handful of studies investigated the direct association between lymphatic transport and atherosclerosis. Most studies extrapolate the findings obtained from the skin. For example, Milasan et al. concluded that improving dermal lymphatic vessels permeability by intradermal apolipoprotein A-I treatment and promoting contraction frequency by systemic administration of VEGF-C before the onset of atherosclerosis in *Ldlr*^−/−^ mice leads to the attenuation of atherosclerosis [123,127]. The above studies did not investigate arterial lymphatic function which would be more relevant to atherosclerosis progression since the function of lymphatic vessels in response to inflammation vary in different tissues.

It is not surprising that assessment of the functionality of the lymphatic vessels in mouse aorta remains scarce. This is largely hampered by insufficient knowledge of peri-vascular lymphatic anatomy and absence of tools to directly inject molecular tracer in small animal like mouse. Inspired by reports published decades ago, our group developed a functional assay to quantitatively measure lymphatic transport in mouse aorta by injecting fluorescent tracer in the adventitia of the mouse abdominal aorta. Transport of macromolecule was taken up by adventitial lymphatics and travelled to iliac-, renal- draining lymph nodes, cisterna chili, and eventually into the thoracic duct in normal non-diseased aorta [86]. To confirm this, deliberate disruption of this draining route was established by ligating iliac efferent and renal afferent lymphatic vessels. The flow of fluorescent tracer was blocked at the point of ligation, resulting the reduction in fluorescent signal detected in the draining lymph nodes. By employing this injection method, we analyzed lymphatic drainage in wild-type and *Apoe^−/−^* mice when lymphangiogenesis is evident and atherosclerosis is well advanced. *Apoe^−/−^* aortic lymphatic exhibited more than 50% reduction in transporting macromolecule and was improved during atherosclerosis and inflammation regression in response to cholesterol-lowering drug treatment [86]. VEGFC-156S treatment enhanced lymphangiogenesis without affecting hypercholesterolemia but did not improve lymphatic drainage, reduce plasma cholesterol, and atherosclerosis lesion size [86]. Expansion of lymphatic vasculature associated with inflammation does not always translate into increased lymphatic drainage in *Apoe^−/−^* mice, hence highlighting the importance of measuring lymphatic functionality to determine the biological significance of lymphangiogenesis in inflammatory diseases. It might also be important to examine the structure of collecting vessels. Defective lymphatic transport in the skin of *Apoe^−/−^* mice is associated with decreased SMC coverage and abnormal valves [126] in collecting vessels and amelioration of these structural changes improves lymphatic function [121]. It would be worth exploring the nature of the extracellular matrix around the collecting lymphatic vessels since build-up of collagen with increasing vascular stiffness may affect pumping efficiency of the collecting lymphatic vessels [128].

## 6. Conclusions

In conclusion, there is little doubt of the essential functional roles of adventitial lymphatic vessels in atherosclerosis including the clearance of immune cells, inflammatory soluble factors, and lipoproteins from the inflamed arterial wall. Present in normal arteries, adventitial lymphatic vessels expand and change their morphology during atherosclerosis (Figure 1). However, lymphatic drainage of macromolecules from the aorta to draining LNs is impaired despite an apparent lymphangiogenesis (Figure 1). This impairment in lymphatic drainage may account for the retention of lipoprotein, inflammatory mediators and cells, and thus participate in adventitial and plaque inflammation (Figure 1). Poor lymphatic drainage may also provide an explanation for the defective inflammation resolution features of atherosclerotic lesions. Notably, lymph stasis was postulated in 1981 by Lemole G to be involved in the development of coronary atherosclerosis based on his observations that mediastinal irradiation was associated with high incidence of coronary artery diseases [129]. This concept was then further supported by recent studies using transgenic mouse models of lymphatic insufficiency crossed with atherogenic mice [130] or atherogenic mice, in which lymphatic drainage was enhanced or, conversely, compromised through surgical ligation [86,92].

Impaired lymphatic function and abnormal morphology seem to be characteristic features of diseases with a chronic inflammatory component. Therefore, restoring the defects of the lymphatic vasculature may be a promising therapeutic target for breaking the pro-inflammatory feedback loop in atherosclerosis and other chronic inflammatory diseases. For this reason, it is urgent to identify novel approaches to therapeutically enhance lymphatic function. Interestingly, specialized pro-resolving mediators also known as bioactive lipids have been shown to play a role in controlling the resolution of inflammation in relation to lymphatic vessels [131,132]. These bioactive lipids may thus serve as a promising potential therapeutic approach to rescue lymphatic function in chronic inflammatory diseases. Finally, elucidation of the mechanisms inducing the morphological and functional alterations in adventitial lymphatic vessels during atherosclerosis may also provide some hint on how to enhance lymphatic function and attenuate the disease.

## Figures and Tables

**Figure 1 cells-10-01344-f001:**
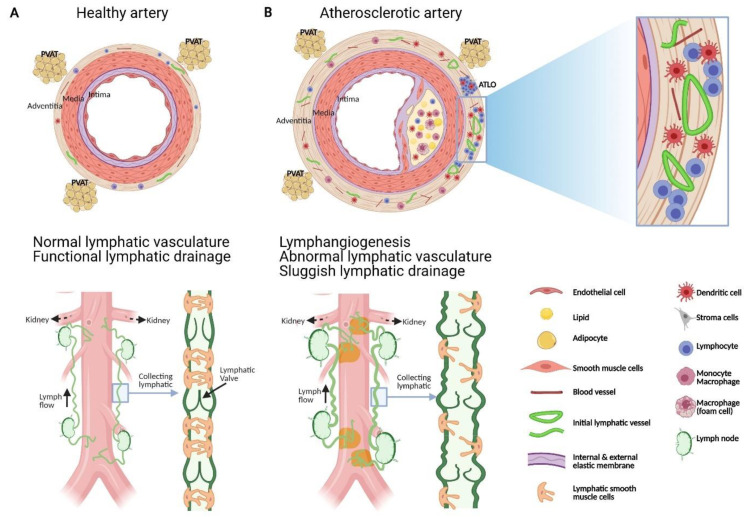
Schematic representation of the structural and functional changes affecting adventitial lymphatic vessels during atherosclerosis. (**A**) Artery is formed by three layers, namely: the tunica intima (inner layer) lined by a monolayer of endothelial cells, the tunica media composed of SMCs, elastin and collagen, and the adventitia, the outer layer formed of fibroblast, collagen, nerve endings, immune cells, blood and lymphatic vessels. Periadventitial adipose tissue (PVAT) is also distributed around arteries. (**B**) Atherosclerosis is characterized by the accumulation of plaques comprising cholesterol, inflammatory soluble factors, immune cells in the intima and inflammation in adventitia that can be associated with the formation of adventitia tertiary lymphoid organ (ATLOs) when the disease is more advanced. (**A**) Similar to lymphatic vessels in other tissues, initial lymphatic vessels in the adventitia converge into larger collecting vessels exhibiting SMC and valves which transport lymph into draining lymph nodes. (**B**) Studies have shown that atherosclerosis is associated with angiogenesis and lymphangiogenesis. Notably, in the abdominal aorta, initial lymphatic vessels appeared more dilated and lymphatic drainage of macromolecule from the adventitia into the draining lymph node is severely compromised during atherosclerosis. Alterations in collecting vessels such as poor SMC coverage, abnormal valve and accumulation of extracellular matrix such as collagen, may account for the impaired lymphatic drainage. These functional defects in adventitial lymphatic vessels may participate to the retention of immune cells, inflammatory cytokines, and lipids in the arterial wall which in turn favour the progression of atherosclerotic plaque, inflammation and ATLOs. Restoring lymphatic drainage in the arterial wall may thus be a promising strategy to treat atherosclerosis.

**Figure 2 cells-10-01344-f002:**
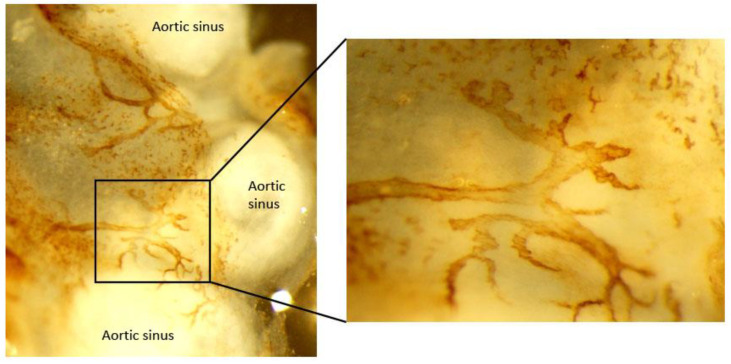
Image of initial lymphatic vessel in adventitia of atherosclerotic mouse aortic sinus. Aorta whole mount was stained for LYVE-1 to identify initial lymphatic vessels in the adventitia of *Apoe*^−/−^ mice.

## Data Availability

Not applicable.

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
