# Peer review of "Leukocyte Trafficking via Lymphatic Vessels in Atherosclerosis"

_cells, 2021, doi:10.3390/cells10061344_

Round 1
Reviewer 1 Report
Very timely review that is nicely illustrated and carefully written by experts in the field. it represents a balanced discussion that highlights both clinical and experimental data sugesting an active role of lymphatics in modulation of atherosclerosis progression.
minor comments
mention of the potential deleterious direct effects of oxLDL on lymphatic endothelial cells (Singla et al 2021, doi:10.3390/antiox10020331)
page 5 line 240: pro-atherogenic AGENTS may enter
page 6 line 277: having one PLEXUS anatomically closer
page 7 line 313: inhibiTION OF VEGFR-3 SIGNALING reduced lymphangiogenesis
figure 2: HEV is not seen in the ATOL structure (too small) so just remove from legend as not mentioned specifically in figure egend either. the media in panel B looks thinner than in panel A
acknowledgement section: missing/ erroneous text inserted
Author Response
Response to Reviewer’s comment
Reviewer 1
Very timely review that is nicely illustrated and carefully written by experts in the field. it represents a balanced discussion that highlights both clinical and experimental data sugesting an active role of lymphatics in modulation of atherosclerosis progression.
minor comments
- mention of the potential deleterious direct effects of oxLDL on lymphatic endothelial cells (Singla et al 2021, doi:10.3390/antiox10020331)
We would like to thank the reviewer to mention this paper. We cited it now line 335.
- page 5 line 240: pro-atherogenic AGENTS may enter. The sentence has been edited accordingly.
- page 6 line 277: having one PLEXUS anatomically closer. The sentence has been edited accordingly.
- page 7 line 313: inhibiTION OF VEGFR-3 SIGNALING reduced lymphangiogenesis. The sentence has been edited accordingly.
- figure 2: HEV is not seen in the ATOL structure (too small) so just remove from legend as not mentioned specifically in figure egend either. the media in panel B looks thinner than in panel A. We agree with the reviewer. Therefore, the HEV has been removed from ATLO in Figure 2. We modified the media thickness in panel B. Please see the edited Figure 2.
- acknowledgement section: missing/ erroneous text inserted. The correct information has been inserted.
Reviewer 2 Report
The manuscript discusses an extremely interesting issue in the context of atherosclerosis and the associated inflammatory response, namely lymphatic system involvement. The lymphatic system is part of the human circulatory system and the immune system and its morphological and functional alterations contribute to the progression of atherosclerosis. Therefore, restoring these defects of the lymphatic vasculature may be a treatment choice for atherosclerosis. This could not be possible without a more advanced knowledge of the mechanisms inducing the morphological and functional alterations in adventitial lymphatic vessels during atherosclerosis.
This manuscript reviews the existing articles in the literature on this topic and discusses all these challenges.
The manuscript is very well organized, wrote and the data are well and clear presented. Figures are also very well designed and suggestive.
Moreover, the manuscript is state-of-the-art, comprehensive and convincing, therefore in this context my recommendation is ‘accept for publication’.
Author Response
Response to Reviewer’s comment
Reviewer 2
The manuscript discusses an extremely interesting issue in the context of atherosclerosis and the associated inflammatory response, namely lymphatic system involvement. The lymphatic system is part of the human circulatory system and the immune system and its morphological and functional alterations contribute to the progression of atherosclerosis. Therefore, restoring these defects of the lymphatic vasculature may be a treatment choice for atherosclerosis. This could not be possible without a more advanced knowledge of the mechanisms inducing the morphological and functional alterations in adventitial lymphatic vessels during atherosclerosis.
This manuscript reviews the existing articles in the literature on this topic and discusses all these challenges.
The manuscript is very well organized, wrote and the data are well and clear presented. Figures are also very well designed and suggestive.
Moreover, the manuscript is state-of-the-art, comprehensive and convincing, therefore in this context my recommendation is ‘accept for publication’.
We thank the reviewer for his positive review.
Reviewer 3 Report
This is a nice and timely review article by an expert in the area. There are a few organizational issues that I note below. There are a significant number of minor English issues throughout (too many to list): a few examples are given below, but there are regular single-plural mismatches, inconsistent use of "the" preceding terms in parallel structure lists, etc. A good readthrough would help.
Odd to throw the first mention of lymphatics/lymphangiogenesis in here at line 145 without defining the native anatomy, lymphangiogenesis, and such as you do later in the paper. The next paragraph (focused on immune accumulation) should precede mention of lymphatics. Lymphatic roles in immune clearance should be defined better before you can say that they might aide in this clearance.
Figure 1 would be much greatly improved it there was an accompanying anatomical diagram showing where this was taken. Really, the Figure 2 cross-sectional diagram of the aorta with normal and diseased state should be referenced earlier. I understand that what is discussed later is necessary to truly appreciate the figure, but illustrating some of the fundamental pathological processes earlier could be helpful.
Line 84: “triggered by the subendothelial retention…” Would not some sort of endothelial activation precede this?
In line 312 where VEGFC is discussed, it might be worth delaying that. The paragraph is about endogenous lymphatic response. The therapy is discussed later.
The paragraph line 318 is oddly thrown in and doesn’t connect the surrounding paragraphs.
Hard to discuss whether the vessels are dysfunctional before defining what roles lymphatics play in athero? Wouldn’t that heading come after laying out their roles in the disease first, then posing the question as to whether dysfunction played a part? This is then heavily re-discussed in the extended conclusions section.
examples of minor text problems:
Line 20: should be medium- and large-sized (again in 64)
Line 57: last past years?
Line 116: statin should be plural
Author Response
Response to Reviewer’s comment
Review 3
This is a nice and timely review article by an expert in the area. There are a few organizational issues that I note below. There are a significant number of minor English issues throughout (too many to list): a few examples are given below, but there are regular single-plural mismatches, inconsistent use of "the" preceding terms in parallel structure lists, etc. A good readthrough would help.
Odd to throw the first mention of lymphatics/lymphangiogenesis in here at line 145 without defining the native anatomy, lymphangiogenesis, and such as you do later in the paper. The next paragraph (focused on immune accumulation) should precede mention of lymphatics. Lymphatic roles in immune clearance should be defined better before you can say that they might aide in this clearance.
We thank the reviewer for his comment and we removed the mention of (lymph)angiogenesis line 145.
Figure 1 would be much greatly improved it there was an accompanying anatomical diagram showing where this was taken. The revised manuscript includes now a lower magnification of the image and the area magnified is boxed to identify better the localization of the lymphatic. The legend has also been edited for more clarity.
Really, the Figure 2 cross-sectional diagram of the aorta with normal and diseased state should be referenced earlier. I understand that what is discussed later is necessary to truly appreciate the figure, but illustrating some of the fundamental pathological processes earlier could be helpful. We agree with the reviewer and Figure 2 is now referenced earlier.
Line 84: “triggered by the subendothelial retention…” Would not some sort of endothelial activation precede this?. This is correct. The sentence has been edited accordingly.
In line 312 where VEGFC is discussed, it might be worth delaying that. The paragraph is about endogenous lymphatic response. The therapy is discussed later. In this sentence, we refer to endogenous VEGF-C expression and not VEGF-C therapy. To clarify this point, we edited the sentence accordingly.
The paragraph line 318 is oddly thrown in and doesn’t connect the surrounding paragraphs. We made a mistake in this sentence. Instead of “ Temporal and spatial assessment of atherosclerosis…” it was supposed to be “Temporal and spatial assessment of lymphangiogenesis”. This has been corrected and therefore the section is connected to the rest.
Hard to discuss whether the vessels are dysfunctional before defining what roles lymphatics play in athero? Wouldn’t that heading come after laying out their roles in the disease first, then posing the question as to whether dysfunction played a part? This is then heavily re-discussed in the extended conclusions section.
We agree with the reviewer’s suggestion and we changed the order of the sections. In the revised manuscript, the section on lymphatic roles in atherosclerosis precedes the section of their potential dysfunction.
examples of minor text problems:
all these minor comments have been addressed as stated below
Line 20: should be medium- and large-sized (again in 64): this has been corrected
Line 57: last past years?. This has been changed to “last years”
Line 116: statin should be plural. This has been modified accordingly.